



**The role of diatom resting spores for pelagic-benthic coupling in the**
**Southern Ocean.**
Mathieu Rembauville[1], Stéphane Blain[1], Clara Manno[3], Geraint Tarling[3], Anu
Thompson[4], George Wolff[4], Ian Salter[1,2*]
[1]Sorbonne Universités, UPMC Univ Paris 06, CNRS, Laboratoire d'Océanographie Microbienne (LOMIC),
Observatoire Océanologique, F-66650, Banyuls/mer, France
[2]Alfred-Wegener-Institute for Polar and Marine research, Bremerhaven, Germany
[3]British Antarctic Survey, Natural Environmental Research Council, High Cross, Madingley Road, Cambridge,
CB3 0ET, United Kingdom.
[4]School of Environmental Sciences, 4 Brownlow Street, University of Liverpool, Liverpool, L69 3GP, United
Kingdom
[*]Corresponding author: ian.salter@obs-banyuls.fr
# Abstract
Natural iron fertilization downstream of Southern Ocean island plateaus support large
phytoplankton blooms and promote carbon export from the mixed layer. In addition to
sequestering atmospheric $CO_2$, the biological carbon pump also supplies organic matter (OM)
to deep-ocean ecosystems. Although the total flux of OM arriving at the seafloor sets the
energy input to the system, the chemical nature of OM is also of significance. However, a
quantitative framework linking ecological flux vectors to OM composition is currently
lacking. In the present study we report the lipid composition of export fluxes collected by
five-moored sediment traps deployed in contrasting productivity regimes of Southern Ocean
island systems (Kerguelen, Crozet and South Georgia) and compile them with quantitative
data on diatom and fecal pellet fluxes. At the three naturally iron fertilized sites, the relative
contribution of labile lipids (mono- and polyunsaturated fatty acids, unsaturated fatty



alcohols) is 2-4 times higher than at low productivity sites. There is a strong attenuation of
labile components as a function of depth, irrespective of productivity.  The three island
systems also display regional characteristics in lipid export.  The diversity of sterols is greater
in the relatively warm waters of the Polar Frontal Zone when compared to the Antarctic zone,
reflecting the transition from mixed phytoplankton communities to principally diatom-derived
OM. An enrichment of zooplankton dietary sterols, such as $C_{27}\Delta^5$, at South Georgia is
consistent with high zooplankton and krill biomass in the region and the importance of fecal
pellets to POC flux. There is a strong association of diatom resting spore fluxes that dominate
productive flux regimes with energy rich unsaturated fatty acids.  At the Kerguelen Plateau
we provide a statistical framework to link seasonal variation in ecological flux vectors and
lipid composition over a complete annual cycle. Our analyses demonstrate that ecological
processes in the upper ocean, e.g. resting spore formation and grazing, not only impact the
magnitude and stoichiometry of the Southern Ocean biological pump, but also regulate the
composition of exported OM and the nature of pelagic-benthic coupling.










## 1. Introduction


The biological pump transfers organic carbon (OC) from photosynthetic production to

the deep ocean (Volk and Hoffert, 1985) with important implications for the sequestration of

atmospheric $CO_2$ (Sarmiento et al., 1988; Kwon et al., 2009). Only a minor fraction of the

carbon fixed in the sunlit ocean reaches the deep ocean and sediments (Martin et al., 1987;

Honjo et al., 2008), but this carbon and energy supply is essential for the functioning of deep-

sea benthic ecosystems (Billett et al., 1983, 2001; Ruhl and Smith, 2004; Ruhl et al., 2008) .

Commonly referred to as pelagic-benthic coupling (Graf, 1989), the composition, lability and

timing of organic matter (OM) flux arriving at the seafloor can exert a large influence on

benthic communities (Billett et al., 2001; Galeron et al., 2001; Mincks et al., 2005; Smith et

al., 2006; Wolff et al, 2011).

Understanding the factors influencing the functioning of the biological pump remains

a central question in biogeochemical oceanography (Boyd and Newton, 1995; Rivkin et al.,

1996; Boyd and Trull, 2007; Guidi et al., 2016). Many different approaches have been

adopted to study the biological pump, including carbon budgets (Emerson et al. 1997,

Emerson 2014), mixed layer nutrient inventories (Eppley and Peterson, 1979; Sarmiento et al.

2004), radionuclide disequilibria (Buesseler et al., 1992; Savoye et al., 2006), optical methods

(Gardner et al., 1990; Guidi et al. 2016), neutrally buoyant- (Buesseler et al. 2000; Salter et al.

2007) and moored-sediment traps (Berger, 1971; Honjo, 1976). Although all of these methods

have their own caveats, sediment traps offer the distinct advantage of collecting and

preserving sinking particles for subsequent biological and chemical analysis. Moored

sediment traps allow the direct quantification of sinking protists including dinoflagellates (e.g.

Harland and Pudsey, 1999), diatoms (e.g. Salter et al. 2012), coccolithophores (e.g. Ziveri et

al. 2007), radiolarians (e.g. Takahashi et al., 1991), silicoflagellates (Rigual-Hernández et al.,



2010), foraminifera (Salter et al. 2014) and zooplankton faecal pellets (Wilson et al., 2008,
2013).  Indirect approaches uses biomarkers such as lipids and amino acids to identify the
source (algal, zooplanktonic, bacterial) and diagenetic status (lability, degree of preservation)
of the exported OM  (Wakeham, 1982; Wakeham et al., 1980, 1984, 1997; Kiriakoulakis et
al., 2001; Wakeham et al., 2009; Lee et al., 2009; Salter et al., 2010). Although it is generally
well-acknowledged that ecological vectors of flux are linked to the geochemical composition,
studies providing a coupled description of biological components and OM composition of
export fluxes remain relatively scarce (e. g. Budge and Parrish, 1998).

Southern Ocean island plateaus such as Kerguelen (Blain et al., 2007), Crozet (Pollard

et al., 2009) and South Georgia (Tarling et al., 2012) provide a natural source of iron to the
iron-poor waters of the Antarctic Circumpolar Current (de Baar et al., 1990; Martin et al.,
1990). Currents and the topography of the sea floor lead to enrichment of iron in waters
adjacent to the islands which supports large diatom-dominated phytoplankton blooms
(Armand et al., 2008; Korb et al., 2008; Quéguiner, 2013) that contrast with the high nutrient,
low chlorophyll (HNLC, Minas et al., 1986) regime that generally prevails in Antarctic
waters. Previous studies of Southern Ocean island plateaus have identified the significance of
resting spore formation by neritic diatom species (*Eucampia antacrica* var. *antarctica*,
*Chaetoceros Hyalochaete*, *Thalassiosira antarctica*) in response to nutrient limitation in mid-
summer (Salter et al., 2012; Rembauville et al., 2015, 2016a). The export of resting spores
generally occurs during short and intense events but they can account for a significant fraction
(40-60 %) of annual carbon flux out of the mixed layer at these naturally fertilized sites. This
process contributes to the ~2 fold increase in annual carbon export when compared to the
HNLC sites (Salter et al., 2012; Rembauville et al., 2015, 2016a).

Despite the general importance of resting spore ecology for POC export from naturally

iron-fertilized systems in the Southern Ocean, there are some notable differences in the nature



of export fluxes from Crozet, Kerguelen and South Georgia. At Crozet, in the Polar Front
Zone (PFZ), the abundance of foraminifers and pteropods leads to a high inorganic to organic
carbon export ratio (1 mol:mol, Salter et al., 2014). At Kerguelen, south of the Polar Front in
the Antarctic Zone (AAZ) the inorganic to organic carbon ratio is much lower (0.07) and
$CaCO_3$ flux is mainly attributed to coccoliths (Rembauville et al., 2016). At South Georgia
(AAZ), the faecal pellet contribution to carbon export is higher (~60 % in summer-autumn
Manno et al., 2015) when compared to Kerguelen (34 % of annual POC flux; Rembauville et
al., 2015). The strong gradients in productivity and ecosystem structure that characterize these
island systems offer a valuable framework to address the link between biological and
geochemical composition of particle export.

The impact of different carbon export vectors on the lability of the exported OM is

necessary to understand the impact of upper ocean ecology for pelagic-benthic coupling (Ruhl
and Smith, 2004; Ruhl et al., 2008). High biomass of meio-, micro- and macrofuna in abyssal
sediments of the Southern Ocean (Brandt et al., 2007) suggests a transfer of OM originating
from photosynthetic autotrophs down to the seafloor. This diversity and biomass is not
geographically homogeneous, but rather constrained by the upper ocean productivity levels
(Wolff et al., 2011; Lins et al., 2015). In this context, the comparison of lipid biomarkers in
export fluxes originating from different sites in the Southern Ocean may help to understand
how ecological processes at the origin of export flux also shape the magnitude and lability of
OM supply to deep-sea benthic communities.

This study compiles lipid biomarker data from five annual sediment trap deployments

in the vicinity of Southern Ocean Island plateaus in order to (i) compare the composition of
lipid biomarkers in export fluxes collected in sites of various productivity levels and across
different depths, (ii) identify how ecological export vectors, in particular resting spores, shape



the lability of POC fluxes over a complete annual cycle and (iii) derive the potential
implications of ecological flux vectors for pelagic-benthic coupling.

## 2 Material and Methods

### 2.1 Trap deployments and sample processing

We compile 5 long-term sediment trap deployments located in the vicinity of island plateaus
in the Southern Ocean (Fig. 1, Table 1). Two sediment traps were located upstream of the
islands in HNLC waters (M6 and P2 at Crozet and South Georgia, respectively) and three
were located in naturally iron-fertilized and productive waters characterized by enhanced
phytoplankton biomass (A3, M5 and P3 at Kerguelen, Crozet and South Georgia,
respectively). The detailed hydrological settings of deployments and bulk chemical analyses
of biogeochemical fluxes have been published previously (Table 1). After the retrieval of each
sediment trap, swimmers (organisms actively entering the trap funnel) were manually
removed from the samples and therefore do not contribute to the lipid fluxes we report.

### 2.2 Lipid analysis

Lipid analyses were performed on 1/8 wet aliquots resulting from the splitting of original
samples. Because of the low amount of material collected in some cups, 1/8 wet aliquots were
combined prior to the lipid analyses (Supplementary Table 1).
Lipids analyses of Crozet sediment trap samples were performed as described by
Kiriakoulakis et al. (2001) and Wolff et al. (2011). For the Kerguelan and South Georgia
samples a similar protocol was used. Briefly, separate 1/8 aliquots were spiked with an
internal standard (5α(H)-cholestane), sonicated (filters; 3 x 15 min;
dichloromethane:methanol 9:1), transmethylated (methanolic acetyl chloride) and silylated



(bistrimethylsilyltrifluoroacetamide; 1 % trimethylsilane chloride; 30–50 µL; 40°C; 0.5–1 h).
GC-MS analyses were carried out using a GC Trace 1300 fitted with a split-splitless injector,
using helium as a carrier gas (2 mL min$^{-1}$) and column DB-5MS (60m x 0.25mm (i.d.), film
thickness 0.1µm, non-polar solution of 5% phenyl and 95% methyl silicone). The GC oven
was programmed after 1min from 60°C to 170°C at 6°C min$^{-1}$, then from 170°C to 315°C at
2.5 °C min$^{-1}$ and held at 315 °C for 15 min. The eluent from the GC was transferred directly
to the electron impact source of a Thermoquest ISQMS single quadrupole mass spectrometer.
Typical operating conditions were: ionisation potential 70 eV; source temperature 215°C; trap
current 300 µA. Mass data were collected at a resolution of 600, cycling every second from
50–600 Thompsons and were processed using Xcalibur software. Compounds were identified
either by comparison of their mass spectra and relative retention indices with those available
in the literature and/or by comparison with authentic standards. Quantitative data were
calculated by comparison of peak areas of the internal standard with those of the compounds
of interest, using the total ion current (TIC) chromatogram. The relative response factors of
the analytes were determined individually for 36 representative fatty acids, sterols and
alkenones using authentic standards. Response factors for analytes where standards were
unavailable were assumed to be identical to those of available compounds of the same class.
**2.3 Statistical analyses**
The lipid composition of sediment trap samples from the five sites was investigated using
principal component analysis (PCA) and the similarity of samples was studied using a
clustering (Ward aggregation criteria) based on lipid classes. This methodology has been used
previously to study the organic geochemistry of sinking particles in the ocean (Xue et al.,
2011). Prior to both PCA and clustering, raw lipid fluxes were transformed by calculating the
square root of their relative abundance within each sample. This transformation followed by
the calculation of the Euclidian distance is also known as the Hellinger distance, which





provides a good compromise between linearity and resolution in ordination analyses
(Legendre and Legendre, 1998; Legendre and Gallagher, 2001).

## 3 Results

### 3.1 Lipid class distribution and seasonality

The total lipid flux collected by sediment traps was five orders of magnitude higher in the
shallow deployment at A3 (230 mg m$^{-2}$ at 289 m), when compared to the deep sediment trap
at M6 (44 µg m$^{-2}$ at 3160 m, Fig. 2, Table 2). The contribution of labile lipids (unsaturated
fatty acids and alcohols, Wolff et al., 2011, Table 2) to total lipid fluxes was 2-4 times higher
in the naturally fertilized sites (20-40 % at A3, P3 and M5) when compared to the HNLC
deployments (<10 % at P2 and M6). Unsaturated fatty acids were dominated (>75 %) by
monounsaturated fatty acids (MUFA) at all sites. Semi-labile lipids (saturated fatty acids
analysed as their methyl esters; FAMEs, and saturated fatty alkanols; Table 2) accounted for a
small fraction (10-15 %) of total lipids at South Georgia, but a higher fraction (30-40 %) at
Crozet. Semi-labile lipids were dominated by the FAME contributions (~70 %) at all sites.
Sterols were the dominant lipids at South Georgia (65-85 %) and were less abundant (30-35
%) at the other sites.
The total lipid flux normalized to OC decreased by four orders of magnitude between
the shallowest (A3, 195.2 mg lipid g OC$^{-1}$) and the deepest (M6, 0.3 mg lipid g OC$^{-1}$)
deployment (Table 2). OC-normalized lipid fluxes in the shallow deployment at Kerguelen
(A3) displayed high contributions from MUFAs (57.7 mg lipid g OC$^{-1}$), PUFAs (13.8 mg
lipid g OC$^{-1}$) and FAMEs (44.6 mg lipid g OC$^{-1}$). All other deployments (P3, P2, M5 and M6)
had much lower amounts of labile and semi-labile compounds and were dominated by sterols
(89.5 – 5 111.5 µg lipid g OC$^{-1}$).





Samples from Crozet (M5 and M6) were positively projected on the first axis of the

PCA together with FAME, $C_{28}$ and $C_{29}$ sterols and long chain unsaturated fatty acids ($C_{22}$,
$C_{24}$) (Fig. 3a). Samples from South Georgia (P3 and P2) were negatively projected on the first
axis, close to $C_{27}$ sterols. Samples from Kerguelen (A3) were positively projected on the
second axis and mainly associated with $C_{16}$-$C_{20}$ unsaturated fatty acids.

Four main clusters of sediment trap samples could be identified based on the largest

distance break after the first node of the dendrogram (Fig. 3b). Cluster A contained most of
the spring and summer samples from the naturally-fertilized sites of Kerguelen and Crozet
(A3 and M5) characterized by the highest relative abundance of labile lipids (PUFA and
MUFA). Cluster B was composed of summer and winter samples from A3 displaying a high
abundance of unsaturated alkenols. Cluster C contained spring and summer samples from the
naturally-fertilized site of South Georgia (P3) and few samples from Kerguelen and Crozet
and were characterized by a mixture of labile, semi-labile and refractory lipids (MUFA,
FAME and sterols). Finally, cluster D was mostly composed of samples from the HNLC site
of South Georgia (P2), displaying a large dominance of sterols.
**3.2 Seasonality at A3**
In spring, vegetative diatoms were the main contributors to the low POC flux, followed by
cylindrical faecal pellets (Fig. 4a). Lipid fluxes were dominated by $C_{16:1}$ (cis-9), hexadecanoic
acid ($C_{16}$ FAME), EPA, (Z)-octadec-9-enoic acid ($C_{18:1}$ (cis-9)), and cholesterol ($C_{27}\Delta^5$) that
altogether contributed to >75% of the total lipids (Fig. 4b). In summer, diatom resting spores
dominated the POC flux, followed by cylindrical and ovoid fecal pellets (Fig. 4c). $C_{16:1}$ (cis-9)
strongly dominated lipid export (47 %), followed by $C_{18:1}$ (cis-9), $C_{27}\Delta^5$ and $C_{29}\Delta^0$ sterols and
EPA (Fig. 4d). In autumn, when tabular faecal pellets dominated the export flux (Fig. 4e),
$C_{27}\Delta^5$ was the major lipid exported followed by $C_{16}$ FAME, $C_{18:1}$ (cis-9), (all Z)-eicosatri-



11,14.17-enoic acid ($C_{20:3}$ (cis-11) and *n*-hexadecanol ($C_{16}$ OH), (Fig. 4f). In winter, large
faecal pellets (tabular and ellipsoid shapes) dominated the carbon flux (Fig. 4g). Dominant
lipids were eicosenol ($C_{20:1}$ OH) and octadecenol ($C_{18:1}$ OH), followed by $C_{16}$ FAME, $C_{27}\Delta^5$
and $C_{18:1}$ (cis-9)), (Fig. 4h).

## 220 4 Discussion

### 221 4.1 Geographical differences in lipid export composition across the
### 222 Southern Ocean island systems

Annual lipid export at the naturally-fertilized sites of Crozet and South Georgia was

characterized by relatively high fluxes of labile and semi-labile compounds compared to the
HNLC sites. Similarly at the iron-fertilized productive site on the Kerguelen Plateau, labile
and semi-labile lipid classes dominate the annual flux profile. The labile lipid class was
dominated by MUFAs, and to a lesser extent, PUFAs.  In particular, two lipid compounds
($C_{16:1}$ (cis-9) and EPA) commonly associated with diatoms (Kates and Volcani, 1966; Lee et
al., 1971) were important components of the labile lipid class. These observations confirm
that the large diatom-dominated phytoplankton blooms observed downstream of island
plateaus (Armand et al., 2008; Korb et al., 2010; Quéguiner, 2013), which are supported by
enhanced iron supply (Blain et al., 2008; Pollard et al., 2009 Nielsdóttir et al., 2012; Bowie et
al., 2015), can result in significant export of labile OM out of the mixed layer.

The PCA and clustering analyses reveal a notable degree of regional structure and

highlight the prevalence of specific lipid classes in the different island systems. The first axis
of the PCA (23.7 % of variance) represents the location of the sediment trap deployments and
the second axis corresponds to the deployment depth. The P3 and P2 sites at South Georgia




both display ~2 times higher relative abundance of sterols compared to the Kerguelen (A3)
and Crozet (M5 and M6) sites. Sterols are important components of the plasma membrane
found in almost all eukaryotic organisms (Dufourc, 2008). Zooplankton use dietary sterols of
phytoplankton origin, preferentially assimilating $C_{27}\Delta^5$, or converting phytosterols to $C_{27}\Delta^5$
(Volkman, 1986, 2003) that are ultimately egested in faecal pellets (Bradshaw and Eglinton,
1993; Prahl et al., 1984). An enrichment in $C_{27}\Delta^5$ (and other $C_{27}$ sterols such as $C_{27}\Delta^{5,22}$ and
$C_{27}\Delta^{22}$) in sinking OM is thus considered indicative of a high contribution of faecal material
(Ternois et al., 1998) to export flux. The relative abundance of $C_{27}\Delta^5$, $C_{27}\Delta^{22}$, $C_{27}\Delta^{5,22}$
compounds is highest in the export fluxes around South Georgia, consistent with the higher
contribution of faecal pellets to carbon export at South Georgia (Manno et al., 2015)
compared to Kerguelen (Rembauville et al., 2015). The biomass of zooplankton groups such
as copepods and pteropods reach some of their highest Southern Ocean abundances in the
northern Scotia Sea, which is also inhabited by Antarctic krill (Ward et al. 2012, Mackey et
al. 2012).
The high contribution of $C_{27}\Delta^5$ and $C_{28}\Delta^{5,22}$ at South Georgia also reflects the dominance of
diatoms at the base of the food web (Korb et al., 2010), whereas the higher contribution of
$C_{29}\Delta^0$ and $C_{29}\Delta^{5,22}$ at Crozet suggests a more diversified phytoplankton community with
possible contributions from Chlorophyceae, Haptophyceae and cyanobacteria (Volkman,
2003; Hernandez-Sanchez et al., 2010, 2012). Warmer waters of the Polar Frontal Zone (PFZ)
at Crozet are known to host a more diversified phytoplankton community compared to the
diatom-dominated waters of the Antarctic one (AAZ) at Kerguelen and South Georgia
(Wright et al., 1996; Fiala et al., 2004; Poulton et al., 2007; Korb et al., 2012; Armand et al.,

2008).

**4.2 Depth-related trends in lipid composition**



The decrease in the total lipid flux of five orders of magnitude between the shallowest
(289 m) and the deepest (>3000 m) deployment is consistent with the trend generally
observed in the global ocean (Wakeham and Lee, 1993; Wakeham et al., 1997, 2009).
Moreover, the strong decrease in OC-normalized lipid flux, particularly in the case of labile
MUFA and PUFA compounds, suggests that lipids are selectively degraded/remineralized
during the sinking of the OM. In the shallowest trap (A3, 289 m), the high OC-normalized
MUFA flux and the abundance of diatom-derived essential PUFA ($C_{16:3}$, $C_{18:6}$, $C_{20:4}$, $C_{20:5}$ and
$C_{22:6}$) reflects the export of fresh and highly labile diatom-derived OM (Dunstan et al., 1993).
By contrast, the presence of branched *iso-* and *anteiso-* $C_{15}$ and $C_{17}$ compounds in the deeper
trap samples may be attributed to the activity of bacterial reworking of the particulate OM in
the deep ocean (Kaneda, 1991; Wakeham et al., 1997).

### 273    4.3 A quantitative framework linking ecological flux vectors to the
### 274    geochemical composition of particles

In order to advance our understanding of the role of ecosystem structure in driving the
composition of particle export, quantitative datasets on both compound and organism fluxes
are required. The dataset from the Kerguelen Plateau was selected as a basis for constructing a
quantitative framework linking ecological flux vectors to the lipid composition of exported
particles. This choice was made primarily on the basis of the high-quality quantitative data on
diatom and fecal pellet fluxes (Rembauville et al. 2015), but also reflects the shifts in
seasonality between dominant flux vectors and highest overall lipid fluxes and concentrations
(Table 2).

### 283    4.3.1 Spring

During spring, the lipid flux is low (0.3 mg m$^{-2}$ d$^{-1}$), as is the corresponding POC flux
(~0.15 mmol m$^{-2}$ d$^{-1}$), which is mainly driven by vegetative diatoms belonging to the genera



*Fragilariopsis*, *Pseudo-nitzschia* and *Thalassionema,* as well as small faecal pellets
(Rembauville et al., 2015). Diatoms are known to predominantly accumulate unsaturated fatty
acids such as $C_{16:1}$ (cis-9), EPA, and to a less extent $C_{18:1}$ (cis-9), (Kates and Volcani, 1966;
Opute, 1974; Chen, 2012; Levitan et al., 2014). Diatoms also produce FAMEs, mainly the $C_{16}$
homologue (Lee et al., 1971; Matsumoto et al., 2009; Liang et al., 2014). Although the lipid
flux is low, the lipid composition we report in spring ($C_{16:1}$ (cis-9), $C_{16}$ FAME, EPA and $C_{18:1}$
(cis-9)) is consistent with a diatom-dominated export assemblage.
**4.3.2 Summer**

Summer at the Kerguelen Plateau is characterized by intense export of diatom resting

spores (*Chaetoceros Hyalochaete* spp. and to a lesser extent *Thalassiosira antarctica)* that
contribute 60% of the annual POC flux (Rembauville et al. 2015) and is associated with the
highest export of total lipids (2.5 mg $m^{-2}$ $d^{-1}$, Supplementary Table 1).  The summer lipid
profile is dominated by $C_{16:1}$ (cis-9)  and $C_{18:1}$ (cis-9), with a marked contribution of EPA.
Higher total lipid contents have been documented in resting spores of *Chaetoceros*
*Hyalochaete* and *Thalassiosira antarctica* when compared to vegetative cells (Doucette and
Fryxell, 1983; Kuwata et al., 1993). Moreover, our results are consistent with the 8-12 fold
increase in the content of $C_{16:1}$ (cis-9) and $C_{18:1}$ (cis-9) in *Chaetoceros pseudocurvisetus*
resting spores when compared to the vegetative stages (Kuwata et al., 1993). An increase in
the cell content of EPA during the formation of resting spores has also been reported for
*Chaetoceros salsugineus* (Zhukova and Aizdaicher, 2001).

Resting spore formation is an ecological strategy utilized by certain diatom species to

persist in environments where unfavorable conditions (e.g. light or nutrient limitation) occur
(Smetacek, 1985; French and Hargraves, 1985; McQuoid and Hobson, 1996). Lipids produce
more energy per unit mass than polysaccharides and are stored in concentrated forms by



diatoms (Obata et al., 2013). The accumulation of energy-rich unsaturated fatty acids in the
resting spore, associated with a reduced metabolism (Oku and Kamatani, 1999) and sinking to
deeper waters (Smetacek, 1985) act in concert to increase the survival rate of the cells. In
order for this ecological strategy to work the cells must be reintroduced to the surface mixed
layer during a period favorable for growth.  Nevertheless, sediment trap studies from Southern
Ocean island systems clearly document that a significant portion of resting spores settle to
depth (Salter et al. 2012, Rembauville et al. 2015, 2016).  Consequently, this ecological
survival strategy of diatoms results in large fluxes of labile lipid compounds arriving at the
seafloor.
Cholesterol ($C_{27}\Delta^5$) was measured (>10 % of total lipids) in settling particles
throughout the year and showed the highest contribution (18 %) in autumn when the
contribution of faecal pellets to POC flux increased. Unlike many eukaryotes, crustaceans are
incapable of *de novo* biosynthesis of sterols and show a simple sterol composition dominated
by $C_{27}\Delta^5$ (Goad, 1981; Baker and Kerr, 1993; Kanazawa, 2001). Its presence throughout the
year can be explained by the continuous export of spherical, ovoid and cylindrical faecal
pellets (Figure 4) which are typically attributed to copepods, amphipods and euphausiids
(Wilson et al., 2008, 2013). Notably we observed the presence of a $C_{29}\Delta^0$ sterol during
summer. $C_{29}$ sterols are abundant in diatoms (Volkman, 2003), and can account for 60 % and
80 % of total lipids of *Navicula* sp., and *Eucampia antarctica var antarctica*, respectively
(Rampen et al., 2010), both of which showed a clear seasonality with a marked summer
maximum (Rembauville et al., 2015).

**4.3.3 Winter**

In winter, the lowest lipid fluxes were recorded and in contrast with other samples was
dominated by mono-unsaturated alkenols ($C_{18:1}$ OH and $C_{20:1}$ OH). These compounds are





generally absent in phytoplankton lipids but are an abundant component in zooplankton wax
ester (Lee et al., 1971), and are often utilized as a marker for zooplankton-derived OM
(Wakeham et al., 1997). More specifically, salp faecal pellets (tabular shape) have been
shown to contain important amounts of $C_{18:1}$ OH and $C_{20:1}$ OH (Matsueda et al., 1986). This is
in good agreement with the dominance of tabular faecal pellets in the winter POC flux at
Kerguelen. Tabular faecal material is present in the export flux during autumn but fatty
alkenols represent a minor constituent of the lipid flux. We expect this difference is primarily
related to the larger contribution of diatoms to export flux (as both single cells or present in
faecal pellets), but it may also reflect changes in zooplankton lipid composition across the
season (Lee et al., 2006). Wax esters are used as energy reserve (Lee et al., 1970) but also
contribute to adjust buoyancy in cold and deep waters in winter (Pond and Tarling, 2011). The
abundance of wax ester-derived compounds we report in winter is also consistent with
observations from neritic areas of the Kerguelen Islands (Mayzaud et al., 2011). Another
indicator of a seasonal shift from diatom (spring) to faecal pellet-dominated export system
(autumn and winter) is the absence of long chain PUFAs in autumn and winter. It has been
previously reported that this energy-rich compound is preferentially assimilated by
zooplankton and is absent in faecal pellets (Stübing et al., 2003).

### 4.4 Implications for pelagic-benthic coupling

Diatom resting spores have been shown to dominate POC flux to the deep ocean in the three
major naturally iron-fertilized island systems of the Southern Ocean (Salter et al., 2012;
Rembauville et al., 2015, 2016a). The present study demonstrates that resting spore flux from
the iron fertilized productive areas around Kerguelen and South Georgia are associated with
higher fluxes of labile MUFA and PUFA lipid classes when compared to nearby HNLC
regimes, comparable to previous findings from the Crozet Plateau (Wolff et al. 2011).. The



oxidation of unsaturated MUFA and PUFA classes produces more energy than their saturated
fatty acid counterparts (Levitan et al., 2014). An energy-rich food supply associated with the
resting spore flux appears to have an important impact on benthic systems. For example, the
decoupling of abundance between megafaunal invertebrates and OM input at Crozet appears
in part to be related to enhanced labile lipid and pigment fluxes supporting higher fecundity of
the dominant megafaunal invertebrate *Peniagone crozeti* (Wolff et al. 2011). At South
Georgia nematode biomass is 10 times higher in deep-sea sediments (>3000m) underlying
iron-fertilized productivity regimes (Lins et al. 2015) whilst OM input varies by considerably
less (Rembauville et al. 2016).  Nematode fatty acids were significantly enriched $C_{16:1}$ (cis-9)
and EPA, two major lipid compounds we have shown to be statistically associated with
summer export events dominated by diatom resting spores.  A resistance to grazing (Kuwata
and Tsuda, 2005) and enhanced sinking velocities or resting spores compared to vegetative
cells (McQuoid and Hobson, 1996) result in their effective transfer to the seafloor
(Rembauville et al. 2016) consistent with the fact they are a common feature of sediments
underlying productive regimes (Crosta et al. 1997; Armand et al. 2008; Tsukazaki et al.
2013). The ecology of resting spore formation therefore acts as an efficient conduit to transfer
energy rich storage lipids to the sediment to such that they play a particularly important role
in pelagic-benthic coupling.
Deep-sea ecosystems are strongly dependent on OM food supply originating from
photosynthesis in the surface ocean (Billett et al., 1983, 2001; Ruhl and Smith, 2004; Ruhl et
al., 2008). In the Southern Ocean, it has been demonstrated that the composition of the upper
ocean plankton community and associated ecological strategies influence the intensity of the
biological carbon pump (Smetacek et al., 2004; Salter et al., 2012; Assmy et al., 2013;
Rembauville et al., 2015) and the carbonate counter pump (Salter et al. 2014).  The present
study demonstrates how changes in major ecological flux vectors, e.g. resting spores versus



fecal pellets, can be linked to the lipid composition of settling particles, with implications for
energy supply to benthic communities.



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





**Tables**
**Table 1:** Information on sediment trap deployments and total fluxes of particulate organic
carbon (POC) and biogenic silica (BSi) collected.

| Location and reference | Sediment trap model | Collection period | Total fluxes (mmol m⁻²) | |
|---|---|---|---|---|
| | | | POC | BSi |
| **Kerguelen** (Rembauville et al., 2015b) | | | | |
| A3 50°38.30' S – 72°02.60' E 289 m | Technicap PPS3/3 0.125 m² | 21/10/2011 – 07/09/2012 No sample lost Total: 322 days | 98 | 114 |
| **South Georgia** (Rembauville et al., 2016a) | | | | |
| P3 52°43.40' S - 40°08.83' W 2000 m | Mclane PARFLUX 0.5 m² | 15/01/2012 – 01/12/2012 1 sample lost Total: 291 days | 41 | 46 |
| P2 55°11.99' S - 41°07.42' W 1500 m | | 15/01/2012 – 01/12/2012 3 samples lost Total: 231 days | 26 | 39 |
| **Crozet** (Salter et al., 2012) | | | | |
| M5 46°00.00' S – 56°05.00' E 3195 m | Mclane PARFLUX 0.5 m² | 28/12/2005 – 29/12/2005 No sample lost Total 360 days | 40 | 165 |
| M6 49°00.03' S – 51°30.59' E 3160 m | | 05/01/2005 – 03/01/2006 No sample lost Total 359 days | 14 | 97 |





**Table 2:** Total annual lipid flux, relative composition of lipid classes and lipid flux
normalized to POC flux for the five sediment trap deployments. Labile - MUFA, PUFA and
Unsat. OH; Semi-labile – FAME, Br. FAME, OH; Refractory – Sterols, Other (Wolff et al.,

750 2011).

| Site | A3 | P3 | P2 | M5 | M6 |
|---|---|---|---|---|---|
| **Total lipid flux** (mg m$^{-2}$ y$^{-1}$) | 230.01 | 3.84 | 2.65 | 1.20 | 0.04 |
| **Relative contribution** (%) | | | | | |
| MUFA | 29.6 | 18.1 | 8.2 | 17.5 | 4.5 |
| PUFA | 7.1 | 1.3 | 0.2 | 3.8 | 0.0 |
| Unsat. OH | 2.3 | 1.1 | 0.6 | 2.1 | 2.2 |
| FAME | 22.9 | 9.7 | 5.9 | 25.9 | 24.7 |
| Br. FAME | 1.4 | 0.2 | 0.3 | 1.0 | 1.1 |
| OH | 8.4 | 2.3 | 1.3 | 13.8 | 15.7 |
| Sterols | 25.9 | 64.9 | 82.3 | 34.7 | 33.7 |
| Other | 2.4 | 2.5 | 1.3 | 1.3 | 18.0 |
| **Normalized total lipid flux** (mg lipid g OC$^{-1}$) | 195.2 | 7.9 | 8.4 | 2.5 | 0.3 |
| **Normalized lipid flux** (µg lipid g OC$^{-1}$) | | | | | |
| MUFA | 57758.2 | 1422.8 | 689.6 | 437.5 | 11.9 |
| PUFA | 13783.6 | 99.7 | 13.0 | 93.8 | 0.0 |
| Unsat. OH | 4431.7 | 83.8 | 47.5 | 52.1 | 6.0 |
| FAME | 44640.5 | 766.6 | 495.0 | 645.8 | 65.5 |
| Br. FAME | 2807.7 | 18.9 | 23.1 | 25.0 | 3.0 |
| OH | 16416.9 | 178.9 | 104.8 | 343.8 | 41.7 |
| Sterols | 50580.6 | 5111.5 | 6878.4 | 864.6 | 89.3 |
| Other | 4769.6 | 199.2 | 107.8 | 31.3 | 47.6 |





**Figures captions.**

**Figure 1:** Location of the five annual sediment trap deployments in the Southern Ocean.
Color refers to surface satellite-derived chlorophyll *a* climatology (MODIS 2002-2016 full
mission product accessed at http://oceancolor.gsfc.nasa.gov/cms/). Dashed and continuous
lines represent respectively the Subantarctic Front (SAF) and Polar Front (PF) from Sallée et
al., 2008. SAZ: Subantarctic Zone, PFZ: Polar Frontal Zone, AAZ: Antarctic Zone.

**Figure 2:** Annual total lipid fluxes (grey bars, left axis) and relative contribution of lipid
classes (coloured bars, right axis) to the total flux from five moored sediment trap
deployments in the Southern Ocean.

**Figure 3:** Association of lipid compounds with sediment trap samples. a) Principal
component analysis of the relative abundance of lipids (n = 121). Black and white symbols
represent respectively the naturally-fertilized and the low productivity sites. b) Clustering of
the sediment trap samples based on the relative abundance of lipid classes (Euclidian distance,
Ward aggregation criteria). Clusters A, B, C and D were defined based on the highest distance
break after the first node. In a) and b), color refers to the lability of lipids according to (Wolff
et al., 2011).

**Figure 4:** Seasonal evolution of carbon export vectors and associated lipid composition over
the central Kerguelen Plateau (A3, 289 m). Left panels: carbon export vectors from
Rembauville et al., 2015. Right panels: sorted relative abundance (coloured bars) and
cumulated relative abundance (dots) of major lipids. a) and b) cups 1-3, c) and d) cup 9, e)
and f) cup 11, g) and h) cup 12.





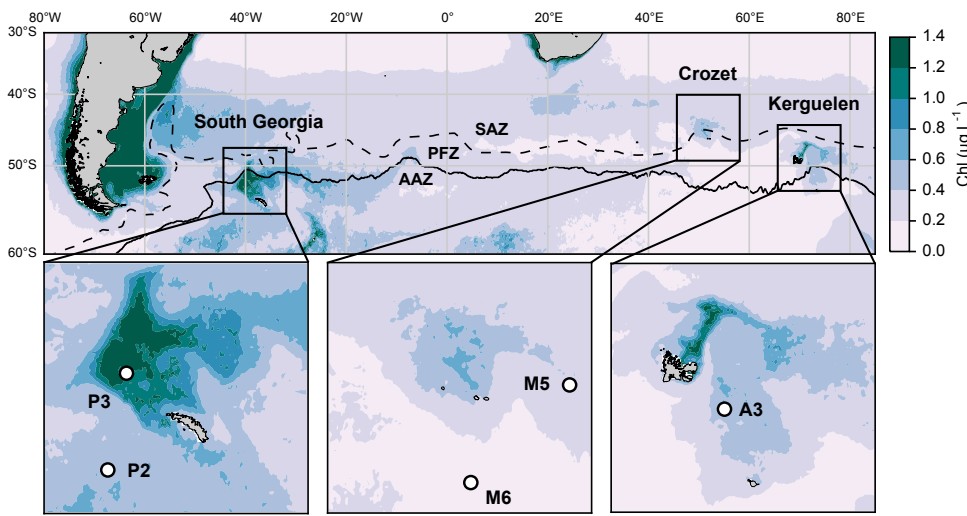


**Figure 1**















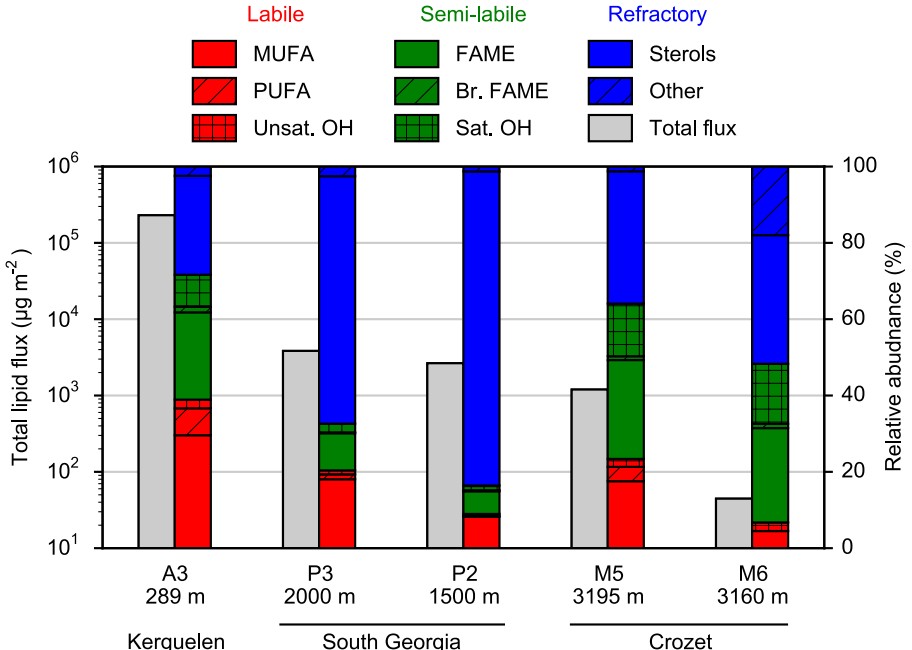


**Figure 2**













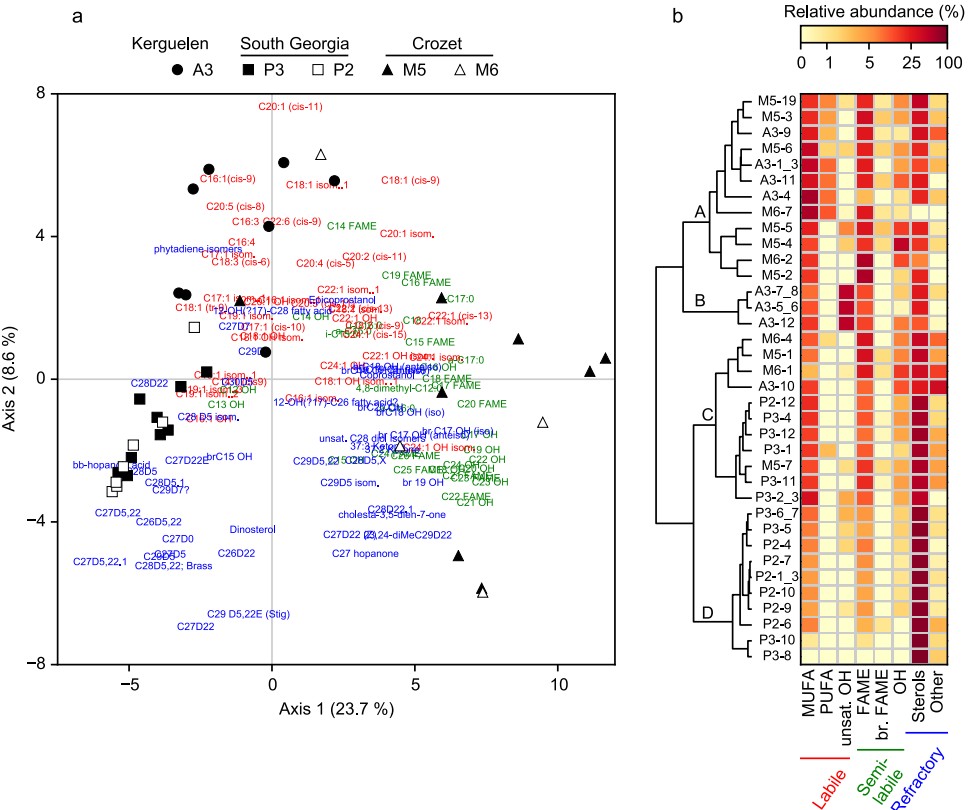


**Figure 3**









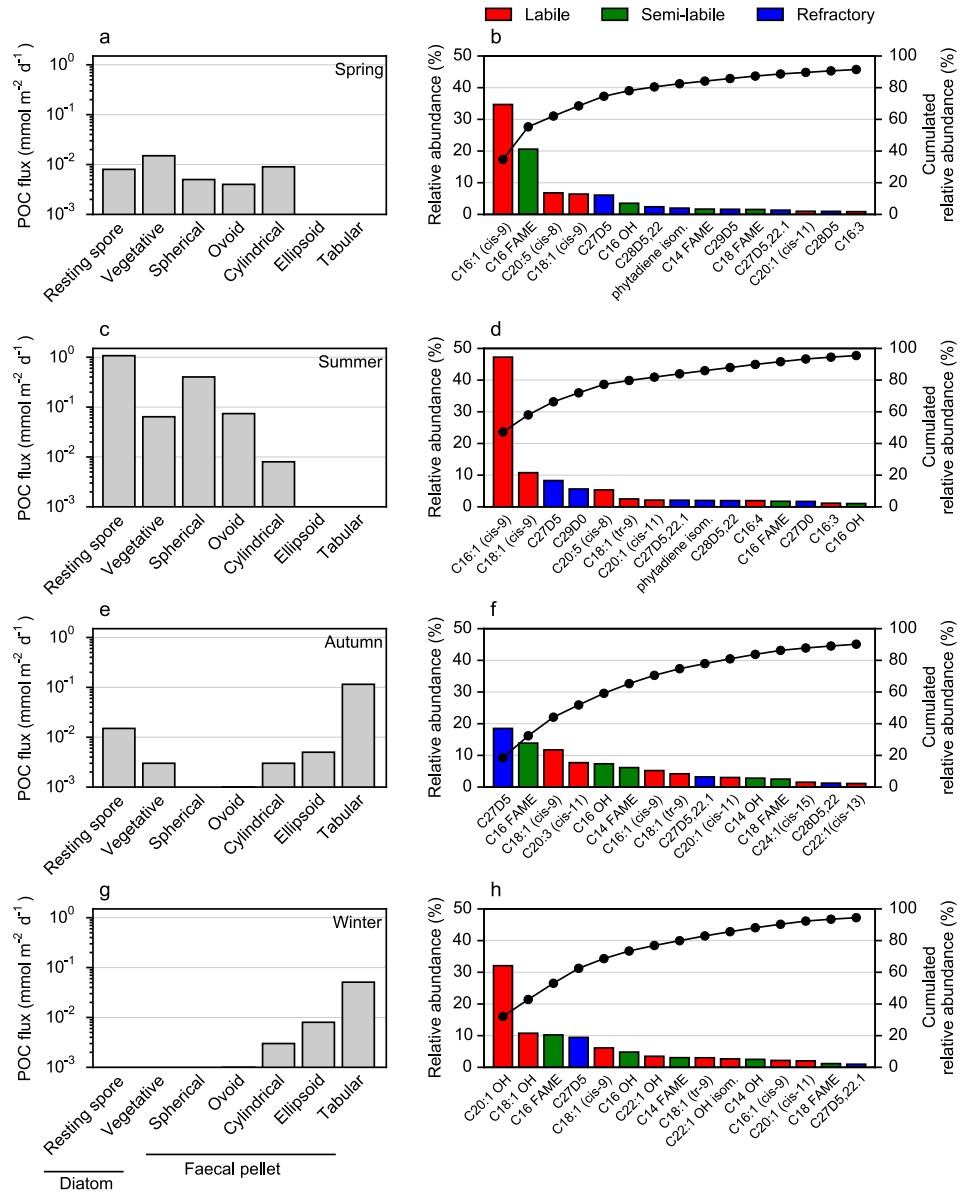


**Figure 4**