# Peer review of "The role of diatom resting spores for pelagic-benthic coupling in the"

_Biogeosciences, 2017_

## Referee Comment (RC1) · Anonymous Referee #1 · 9 Dec 2017

The paper reports on lipid composition in five sediment traps placed in different sites of the Southern Ocean, characterized by different productivity: two of them (M6 and P2) were in HNLC waters, while three other sites (A3, M5 and P3) were located in naturally iron-fertilized areas characterized by higher productivity. Lipid composition markedly differed among these sites, with higher proportion of labile lipids in the naturally iron-fertilized waters. For one of the sediment traps (A3 located in the Kerguelen Plateau), quantitative data on the composition of settled material were available from a previous study. These data have been used to depict the seasonal trend in lipid composition as related to the biological components (diatom cells, resting spores and faecal pellets). Samples collected during the summer period were dominated by diatom resting spores, which transferred to depth a considerable amount of lipids, dominated by the labile

[Figure]

MUFA and PUFA.

The study is interesting since provides the link between qualitative composition of sediment traps and the composition in lipids, which are also used as biomarkers to infer the origin of sinking material.

I have the following comment: The comparison between trap content and lipids was done for the sediment trap A3, which was deployed at relatively shallow depth as compared to the other traps. It is mentioned that the labile lipids can be degraded/remineralized with depth. Can the 'fingerprint' of lipids derived from diatom spores be preserved in the deeper layers? The other sediment traps placed in iron-enriched areas (P3 and M5) were much deeper; is it possible to state that the composition of lipids in these deep stations still reflects the contribution of diatom spores? Or of diatoms in general? The role of diatom spores in mediating a considerable carbon flux for the benthic organism has been demonstrated for the shallower areas: can it be extended to the deep stations in the productive areas as well? I would suggest the points listed above be addressed in the discussion.

---

## Referee Comment (RC2) · Anonymous Referee #2 · 14 Dec 2017

This manuscript by Rembuaville et al describes lipid fluxes and compositions for five deep-water sediment traps deployed in the Southern Ocean to evaluate relationships between Fe fertilization and organic matter (OM) flux to the seafloor and the effects of seasonality in upper ocean ecology and particle morphologies on the ON flux. The major conclusions are that labile lipid fluxes are higher at the three iron-fertilized sites that at the two lower productivity HNLC sites, that lipid fluxes are greatly attenuated with depth, and that specifics of lipid flux and composition depend on ecological factors in the upper ocean. The conclusion that diatom resting spores and zooplankton grazing on the main phytoplankton that produces fecal pellets of various morphologies are major determinants on OM and lipid flux to the seafloor at these sites, and specifics of the fluxes and types of particles varies between the site. Unfortunately there are some

serious flaws with the present form of the manuscript that would require attention of the authors before proceeding further. First and foremost, the lipid compositional data are actually given short shrift. Are they supposed to be in Supplemental table 1 (noted at line 137 and again at line 297), which apparently was either not included (intended or otherwise?) or simply not available to this reviewer. This would actually include the seasonal data for the plot shown in Fig 4. Without a more complete data set (indeed at line 276 "quantitative datasets on both compound and organisms fluxes are required" tells all). The lipid compositional data that are included in the text are too sparse for the generalized conclusions that are drawn. Throughout the text, and in Table 2, there is some confusion about fluxes vs concentration. E.g., at line 284, the "spring lipid flux is low (0.3 mg/m2). . .", but this is a rate not a flux without some time frame. Is it supposed to be per day, or over the deployment period, or annualized? Likewise, in Table 2, the "normalized total lipid fluxes, mg lipid/g OC" are concentrations not fluxes (it is correct that lipid flux/OC flux will give a concentration of mg lipid/g OC). (Note that in figs 2a, c, g, and e fluxes have correct units.) Other comments: P 6 - were the traps treated with some sort of poison or preservative to preserve the integrity of the OM over the entire deployment period (if this is given in one of the references it was not clear)? Line 173 - this is the first instance of the flux "units" being incorrect. See also line 185 – a concentration. . .. What are the annualized primary productivity levels at the five sites? And annualized chl-a concentrations (is that what's in fig 1?). This information is important to give some sense of the quality of the OM produced at each site, as this impacts the comparative fluxes near the seafloor. Line 187- what are monounsaturated FAMES MUFAs, and polyunsaturated FAMES PUFA, but saturated FAMES are FAMES (why not Sat-FAMES or SFMES) for consistence. After all, everything measured is a FAME. Line 201 - isn't "unsaturated alkenols" redundant since alkenols are unsaturated by definition? Line 209 - maybe spell out the name of EPA the first time it is used. Line 252 - why does the cholesterol reflect the dominance of diatoms are the base of the food web? The references cited later (e.g., Rampen et al., 2010; line 328) shows that C27-5 is a relatively minor component of many diatoms. What about 24-methylene

cholesterol (C28-5,24(28)? It's hard to evaluate these statements without the data, which as noted above are missing for some reason. See also paragraph at line 326. Line 266 – The conclusion that lipids are selectively degraded/remineralized during sinking relative to OC because lipid/OC ratios decrease between the shallower and the deepest traps also depends in part on the lipid/OC ratios of the starting material. Since there are apparently differences in production, or chl-a, or ecology between the Fe-fertilized sites and the HNLC sites, what are lipid/OC ratios in the material that is initially exported out of the surface waters? Fig. 2. Are the units on the abscissa supposed to be fluxes. Under "refractory", what is the "other" category? Table 2 – are POC and BSi annual "fluxes" with flux units? There is no discussion of BSi in the manuscript.
* * *

---

## Author Comment (AC1) · 13 Feb 2018

We would like to thank the reviewer for taking the time to carefully read and comment on our manuscript. Below is a point-by-point response to the questions and comments raised.

The paper reports on lipid composition in five sediment traps placed in different sites of the Southern Ocean, characterized by different productivity: two of them (M6 and P2) were in HNLC waters, while three other sites (A3, M5 and P3) were located in naturally iron-fertilized areas characterized by higher productivity. Lipid composition markedly differed among these sites, with higher proportion of labile lipids in the naturally iron-fertilized waters. For one of the sediment traps (A3 located in the Kerguelen Plateau),

quantitative data on the composition of settled material were available from a previous study. These data have been used to depict the seasonal trend in lipid composition as related to the biological components (diatom cells, resting spores and faecal pellets). Samples collected during the summer period were dominated by diatom resting spores, which transferred to depth a considerable amount of lipids, dominated by the labile The study is interesting since provides the link between qualitative composition of sediment traps and the composition in lipids, which are also used as biomarkers to infer the origin of sinking material.

R1 – Q/C – 1: I have the following comment: The comparison between trap content and lipids was done for the sediment trap A3, which was deployed at relatively shallow depth as compared to the other traps. It is mentioned that the labile lipids can be degraded/ remineralized with depth. Can the 'fingerprint' of lipids derived from diatom spores be preserved in the deeper layers? The other sediment traps placed in iron-enriched areas (P3 and M5) were much deeper; is it possible to state that the composition of lipids in these deep stations still reflects the contribution of diatom spores? Or of diatoms in general? The role of diatom spores in mediating a considerable carbon flux for the benthic organism has been demonstrated for the shallower areas: can it be extended to the deep stations in the productive areas as well? I would suggest the points listed above be addressed in the discussion.

R1 – R – 1: We would like to thank the reviewer for this comment. To clarify, we have available detailed diatom counts and lipid analyses for all of the sediment trap samples, including the deeper stations P3 and M5. In the current manuscript, we thus aim to compare the trap content with lipid composition for all the samples.

We know from our previous work that diatom-resting spores account for 60% of annual carbon export from at 300m from an iron-fertilized bloom on the Kerguelen plateau (Rembaubille et al. 2015). We have observed similar patterns in deep samples (>1500m) from the productive regime at South Georgia (P3), whereby 42% of annual carbon export could be attributed to resting spores (Rembauville et al. 2016). At the

productive Crozet site (M5), Eucampia antarctica resting spores dominate flux in the bathypelagic and are strongly correlated with total POC flux (Salter et al. 2012). This is in stark contrast to the HNLC sites from these areas, which have very few resting spores with a negligible contribution to organic carbon flux. These findings, from different island systems, provide strong evidence that diatom flux, in particular resting spores of Chaetoceros and Eucampia, are the dominant vector of organic carbon flux to the bathypelagic (>1500m) ocean following iron fertilized blooms. In the present manuscript we are also able to demonstrate that these resting spore-dominated systems not only transfer significant amounts or organic carbon to the deep ocean, but also mediate a bathypelagic flux of labile lipids in the form of mono- and polyunsaturated fatty acids. For example, consulting Table 2 and Figure 2 in the manuscript, it is clear that the relative abundance and concentration of MUFAs and PUFAs is considerably higher in the productive sites, when compared to the HNLC sites (cf M5 and M6; P3 and P2). All of these samples are >1500m in depth. We thus take this as strong evidence that the signature of enhanced unsaturated fatty acids associated with resting spores is transferred to the bathypelagic ocean.

We acknowledge that this was perhaps not stated as explicitly as it might have been. We have rewritten the first paragraph of section 4.4 (Implications for pelagic-benthic coupling) in order to express these considerations more thoroughly (Lines 363-378).

Please Note: Revised manuscript with track changes, responses to all reviewers, and supplementary information is attached as a compressed file.

Please also note the supplement to this comment:
https://www.biogeosciences-discuss.net/bg-2017-414/bg-2017-414-AC1-supplement.zip

---

## Author Comment (AC2) · 13 Feb 2018

We would like to thank the reviewer for taking the time to carefully read and comment on our manuscript. Below is a point-by-point response to the questions and comments raised.

R2 – Q/C – 1: This manuscript by Rembuaville et al describes lipid fluxes and compositions for five deep-water sediment traps deployed in the Southern Ocean to evaluate relationships between Fe fertilization and organic matter (OM) flux to the seafloor and the effects of seasonality in upper ocean ecology and particle morphologies on the ON flux. The major conclusions are that labile lipid fluxes are higher at the three iron-fertilized sites that at the two lower productivity HNLC sites, that lipid fluxes are

greatly attenuated with depth, and that specifics of lipid flux and composition depend on ecological factors in the upper ocean. The conclusion that diatom resting spores and zooplankton grazing on the main phytoplankton that produces fecal pellets of various morphologies are major determinants on OM and lipid flux to the seafloor at these sites, and specifics of the fluxes and types of particles varies between the site. Unfortunately there are some serious flaws with the present form of the manuscript that would require attention of the authors before proceeding further.

R2 – R – 1: We thank the reviewer for the precise synopsis of our manuscript. In the detailed comments below we aim to address the issues concerning the reviewer.

R2 – Q/C – 2: First and foremost, the lipid compositional data are actually given short shrift. Are they supposed to be in Supplemental table 1 (noted at line 137 and again at line 297), which apparently was either not included (intended or otherwise?) or simply not available to this reviewer. This would actually include the seasonal data for the plot shown in Fig 4.

R2 – R – 2: We would like to assure the reviewer that we would not reference a supplementary file containing the lipid compositional data and then intentionally exclude it from the manuscript. It seems for some reason the reviewer did not receive the supplementary tables, most probably due to an error during the submission process. The lipid data is highly complex: it is a matrix containing 2832 elements that clearly precludes a detailed discussion of all the patterns observed. We have attempted to present and discuss the lipid data we consider most pertinent to our objectives of examining the link between ecological flux vectors and organic matter composition. We agree with the reviewer that this data should be available to validate the patterns we describe, which is why we intended to present it as supplementary information. We also recognize the some of the detail in the patterns of specific compounds we don't fully describe in the manuscript might be of interest to a certain sector of the readership. As originally intended, tables of lipid fluxes and concentrations are presented as Supplementary Tables 1-5 in the revised version of the manuscript.

R2 – Q/C – 3: Without a more complete data set (indeed at line 276 "quantitative datasets on both compound and organisms fluxes are required" tells all).

R2 – R – 3: This comment is somewhat confusing. At line 276 we make a general statement that quantitative flux datasets are required for both chemical elements (in this case lipids) and biological elements (diatoms, fecal pellets etc) in order to provide a quantitative framework and examine mechanistic processes related to pelagic-benthic coupling. By "referencing a complete dataset" we assume the reviewer is revisiting their previous point concerning the lipid compositional data. As stated above, the intention was to provide this complete dataset and it is available as part of our revised submission.

R2 – Q/C – 4: The lipid compositional data that are included in the text are too sparse for the generalized conclusions that are drawn.

R2 – R – 4: We have added more quantitative references to the text, in addition to providing full supplementary tables of lipid compositional data that may be used as a reference supporting our general conclusions.

R2 – Q/C – 5: Throughout the text, and in Table 2, there is some confusion about fluxes vs concentration. E.g., at line 284, the "spring lipid flux is low (0.3 mg/m2) ... ", but this is a rate not a flux without some time frame. Is it supposed to be per day, or over the deployment period, or annualized?

R2 – R – 5 We acknowledge that there might be some confusing terminology throughout the manuscript, which we have tried to address in the revised version (see below). It is perhaps worth clarifying that a rate, by definition, describes some change in a property as a function of time. We would therefore politely disagree with the reviewer that 0.3 mg/m2 would be considered a rate, since it lacks a time component. A flux, on the other-hand, is a special kind of rate that describes the passage of some property through a defined area, integrated over a certain time interval. In sediment trap studies, it is sometimes the case that the time interval over which fluxes are integrated

corresponds to the sediment trap deployment period (which is of course a unit of time), in which case it is entirely correct to write xx mg/m2 over the deployment period as a flux, if the time period is specified. The collection periods for the sediment traps are of course known and were presented in Table 1 of the original version of the manuscript.

The reviewer is specifically referencing our use of the term flux at line 284. However, in the submitted version of the manuscript the units at line 284 are 0.3 mg m-2 d-1 (i.e. including the time component), not 0.3 mg m-2 (excluding the time component) stated by the reviewer, who appears to have misquoted our original text.

R2 – Q/C – 6: At line Likewise, in Table 2, the "normalized total lipid fluxes, mg lipid/g OC" are concentrations not fluxes (it is correct that lipid flux/OC flux will give a concentration of mg lipid/g OC). (Note that in figs 2a, c, g, and e fluxes have correct units.)

R2 – Q/C – 6: The reviewer has a valid point here concerning terminology. It is of course the case that normalizing lipid fluxes to organic carbon fluxes removes both the units of area and time, resulting in two mass quantities expressed as a ratio that can also therefore be considered as a concentration. In the original version of the manuscript we had chosen the terminology "normalized lipid flux", but in order to avoid unnecessary confusion we follow the reviewers suggestion and have changed this to concentration throughout the manuscript.

R2 – Q/C – 7: Other comments: P 6 - were the traps treated with some sort of poison or preservative to preserve the integrity of the OM over the entire deployment period (if this is given in one of the references it was not clear)?

R2 – R – 7: Yes, the detailed processing of samples is in the references in Table 1. This has now been referenced in this section.

R2 – Q/C – 8 Line 173 - this is the first instance of the flux "units" being incorrect. See also line 185– a concentration ...

R2 – R – 8. @line 173 - As explained above these are fluxes integrated over the

deployment period. In the original text this was phrased as "The total lipid flux…" However, this is clearly causing some confusion and so the text has been modified as follows:

Total lipid fluxes integrated over the sediment trap deployment period (Table 1.)

@line 185 – As stated above we have changed our terminology from normalized lipid flux to concentration.

R2- Q/C – 9: What are the annualized primary productivity levels at the five sites? And annualized chl-a concentrations (is that what's in fig 1?). This information is important to give some sense of the quality of the OM produced at each site, as this impacts the comparative fluxes near the seafloor.

R2 – R – 9: The chlorophyll maps presented in Figure 1 show the annual climatologies of MODIS-chl data at each site. This information has been added to the figure legend. The purpose of presenting these maps was primarily to show the different biomass regimes at the iron-fertilised and "HNLC" sites where the sediment traps were deployed. We do not have a good estimate of primary productivity levels at each site, as these would depend on using satellite data and a model to convert chlorophyll to productivity. Such algorithms are not well validated in the Southern Ocean. Furthermore, in order to achieve this one would need to estimate the surface catchment area over which the sediment traps are integrating based on particle trajectories. In order to compute particle trajectories one would need a good estimation of particle sinking speed and vertically resolved current vectors. None of these variables are well constrained at our deployment sites.

We are also not clear that annualized primary productivity levels would offer much insight concerning the quality of OM produced at each site. OM quality is related to the geochemical composition of OM. The reviewer appears to be suggesting that there might be some relationship between the magnitude of organic matter production and it's lipid composition. We are not aware of any studies supporting this notion.

R2- Q/C – 10: Line 187- what are monounsaturated FAMES MUFAs, and polyunsaturated FAMES PUFA, but saturated FAMES are FAMES (why not Sat-FAMES or SFMES) for consistence. After all, everything measured is a FAME.

R2- R – 10: I am not sure we used the terminology described above. At line 187 we say: "….displayed high contributions from MUFAs (57.7 mg lipid g OC-1), PUFAs (13.8 mg lipid g OC-1) and FAMEs (44.6 mg lipid g OC-1)"

MUFAs are monounsaturated fatty acids (1 double bond) PUFAs are polyunsaturated fatty acids (>1 double bond) FAMES are saturated fatty acids.

The reviewer is correct that it is unnecessary to include the ME suffix for FAMES, although this nomenclature is sometimes encountered in the litereature. However, we have followed the reviewer's suggestion and changed to FAMES to saturated fatty acids throughout the manuscript.

R2- Q/C – 11: Line 201 - isn't "unsaturated alkenols" redundant since alkenols are unsaturated by definition?

R2- R – 11: Yes, this is true. I think in a previous version we had used the terms saturated and unsaturated alcohols, the latter of course also being an alkenol. Somehow we changed the terminology to alkenol but unsaturated was not removed. Thank your for picking up on this mistake. We have corrected it throughout.

R2- Q/C – 12: Line 209 - maybe spell out the name of EPA the first time it is used.

R2- R – 12: Done.

R2- Q/C – 13: Line 252 - why does the cholesterol reflect the dominance of diatoms are the base of the food web? The references cited later (e.g., Rampen et al., 2010; line 328) shows that C27-5 is a relatively minor component of many diatoms. What about 24-methylene C2 cholesterol (C28-5,24(28)? It's hard to evaluate these statements without the data, which as noted above are missing for some reason. See also paragraph at line 326.

R2- R – 13: This was a typing mistake, C27Δ5, should have been written as C27Δ5,22, which appears to be more prevalent in Bacillariophycaea (Volkman, 2003). However, upon further reflection we have decided to remove the paragraph that links patterns in sterols to phytoplankton composition. There is too much ambiguity as many sterols are not completely diagnostic as they occur in different groups of phytoplankton.

R2- Q/C – 14: Line 266 – The conclusion that lipids are selectively degraded/remineralized during sinking relative to OC because lipid/OC ratios decrease between the shallower and the deepest traps also depends in part on the lipid/OC ratios of the starting material. Since there are apparently differences in production, or chl-a, or ecology between the Fe-fertilized sites and the HNLC sites, what are lipid/OC ratios in the material that is initially exported out of the surface waters?

R2- R – 14: We agree with the reviewer that the composition of organic matter produced had not been explicitly mentioned. The reason being that we do not have time-series, or point measurements, of lipid/OC ratios in organic matter produced in the mixed layer that could be sensibly compared with our sediment trap measurements. As mentioned above we are not familiar with reports that changes in productivity or biomass are linked to lipid/OC ratios directly, although they may of course be linked indirectly due to patterns related to the taxonomic composition of plankton under different productivity regimes. However, we would not be able to draw any strong inferences on diatom/plankton community structure from productivity estimates and would be therefore unable to link these to variability in lipid/OC concentrations from different bloom areas. As stated in the text, the lipid flux decreases by 5 orders of magnitude between the shallowest and deep trap, which is accompanied by a loss of labile lipid classes such as MUFAs and PUFAs. These features are entirely consistent with degradation over depth. However, we now mention in the manuscript that part of the difference observed at different depths may be related in part to that lipid/OC of "starting material"

Line 331-333: It is possible that some of the differences observed over depth may be related to the initial lipid composition of organic material produced in the photic zone

by different phytoplankton taxa

R2- Q/C – 15 Fig. 2. Are the units on the abscissa supposed to be fluxes. Under "refractory", what is the "other" category? Table 2 – are POC and BSi annual "fluxes" with flux units? There is no discussion of BSi in the manuscript.

R2- R – 15: Yes these are fluxes integrated over the sediment trap deployment period (c.f. R2-R-5). We have added some extra information to the Figure caption to clarify this point. Line 962-963: Total lipid fluxes (grey bars, left axis) integrated over the sediment trap deployment periods (Table 1)......

The "other" category is all of the other lipid compounds that were measured but do not fall into any of the listed categories. A full list of these compounds is available in the supplementary files and these files have now been referenced in the figure caption. We have also made the same correction for the table captions where the category "other" appears.

Yes the fluxes in Table 1 are integrated over the sediment trap deployment period. The table caption has been modified to clarify this point.

Line 293-294: Information on sediment trap deployments and fluxes of particulate organic carbon (POC) integrated over the deployment period.

Biogenic silica data has been removed.

Please Note: The revised manuscript with track changes, responses to all reviewers comments and the supplementary tables are included as a compressed file.

Please also note the supplement to this comment:
https://www.biogeosciences-discuss.net/bg-2017-414/bg-2017-414-AC2-supplement.zip